# OpenReview forum: "ShieldAgent: Shielding Agents via Verifiable Safety Policy Reasoning"
_ICML.cc/2025/Conference — ICML 2025 poster_

### Official Review · Reviewer_GLxp · 2025-02-17

**Overall Recommendation:** 4

**Summary:**

The paper proposes a guardrail/shielding scaffold for LLM-based agents. The core idea is to parse safety documentation such as policies into atomic rules, which are then encoded as probabilistic circuits. The use of probabilistic circuits thus allows for efficient marginal inference of the likelihood that a rule will be violated given some action being taken by the agent. In addition to this main contribution, the paper also contributes an extensive benchmark containing over a thousand examples of unsafe agent trajectories. Empirically, the method proposed in the paper achieves higher accuracies (in terms of predicting whether an action is safe or not, and which rule it is that will be violated) compared to previous work.

## update after rebuttal
I have kept my original score of 4 - Accept. I believe the additional experiments and discussion provided by the authors during the rebuttal, including an additional set of ablations, to be very valuable; my reason for not raising my score further is that this is not my area of expertise, and other reviewers, in particular tHcj, have voiced concerns that appear to be valid. I am not sure I share them, for example I do not find the overall framework to be unclear, but nonetheless the concerns of the other reviewers should be taken into account when making the final recommendation.

**Claims And Evidence:**

I believe the claims to be clear and well supported by the given evidence. However, I would have liked to see more discussion of the impact of the hyperparameters, of which there appears to be many (e.g. the rule similarity threshold $\theta$ and $\epsilon_\text{tol}$), as this appears to be a crucial part of trading off the accuracy and runtime of the safety harness.

**Essential References Not Discussed:**

None to me knowledge.

**Experimental Designs Or Analyses:**

The experiments appear to be both sound and thorough. The benchmarks aren't huge, but given the complexity of constructing such datasets this does not feel like a fair criticism of the work, especially as the authors already supplement their main methodological contribution with a novel benchmark themselves.

**Methods And Evaluation Criteria:**

Yes, the methods and criteria makes sense. The additional contribution of a new benchmark for this task strengthens the paper (since to be honest there is not--to the extent of my knowledge--many existing benchmarks out there to evaluate these methods on), and it does not appear to be constructed in a way that would unfairly favor the author's proposed approach.

My only critique in the terms of the criteria is that in the context of ensuring safety, false negatives are much worse than false positives. I would thus have liked to see an evaluation in terms of the false negative rate, in addition to the accuracy and false positive rate.

**Other Comments Or Suggestions:**

There is a minor type on line 191.

**Other Strengths And Weaknesses:**

The paper is very clearly written. While the methodology is not particularly technically involved, it does have a lot of moving parts, and yet they are presented in a way that (in my opinion) is very easy to follow. Besides that, I applaud the authors for focusing their efforts on an issue which will be very important as agentic systems are likely to become more and more popular over the next few years.

**Questions For Authors:**

1. Have you carried out an ablation for the different parts of your system; for example, how often does your system fail because of a poorly tuned probabilistic circuit vs. a misformulated rule?
2. How do you set the hyperparameters for your method? (I didn't see any discussion of this in the appendix, but please correct me if I am wrong.)

**Relation To Broader Scientific Literature:**

The paper appears to be a significant contribution to the agent literature. Not only is this type of work very important, but the methodological decisions are themselves very interesting and (to my knowledge) novel, such as the use of probabilistic circuits to make exact safety inference tractable. Previous works which I am familiar with, such as ShieldLM by Zhang et al. (2024), were much more limited in scope and largely relied on the LLM to implicitly evaluate the risk of a rule violation; in particular, framing the guardrail problem as an MDP and using exact probabilistic inference to compute the probability of a rule being violated appears to be a much more well-founded approach to the problem.

**Theoretical Claims:**

N/A; no theoretical claims made.

---

> ### Author Rebuttal · Authors · 2025-04-01
>
> We thank the reviewer for recommending our paper for acceptance and we really appreciate their valuable suggestions! Below we have addressed the questions one by one.
>
> > more discussion of the impact of the hyperparameters
>
> Following the reviewer's suggestions, we have provided a detailed ablation study regarding the hyperparameters, i.e., different components, different PC weights, and different safety thresholds $\theta$ in Table-reb 2, 3, 4 below. The results show that guardrail performance and efficiency are primarily influenced by tool calling and model-checking components, which are critical for detecting rule violations. Additionally, we observe that PC significantly reduces the FPR and remains stable even under large perturbations, demonstrating the robustness of our safety certification algorithm.
>
> Table-reb 2: Ablation study on different components
>
> | Component           | ACC@G | FPR@G | FNR@G | ARR@R | Time |
> |------------------|:-----:|:-----:|:-----:|:-----:|:-----:|
> | w/o Model checking     |85.4      |  7.2     | 9.0    | 80.9      | 24.0 |
> | w/o PC     | 82.5      |  13.0     | 2.3    | 87.5      | 31.1      |
> | w/o History cache | 90.4      |  5.6      | 6.5   | 87.5      | 42.0     |
> | w/o Workflow memory  | 86.0      |  7.0      | 8.5   | 82.0      | 38.2     |
> | w/o Tool calling          | 81.4      | 6.2      | 11.3      |  71.3     | 48.9     |
>
> Table-reb 3: Ablation study on PC weights.
>
> | Method           | ACC@G | FPR@G | FNR@G | ARR@R |
> |------------------|:-----:|:-----:|:-----:|:-----:|
> | Learn from real data      | 90.4      |  5.6      | 6.5   | 87.5      |
> | Pseudo-learning |  88.0     |  6.7     | 8.5    | 87.5      |
> | FOL ($\theta \to \infty$)     | 82.5      |  13.0     | 2.3    | 87.5      |
> | Perturbed ($\epsilon_\theta=10\%$)     |  89.7     |  5.9     |  7.7   |   87.5    |
> | Perturbed ($\epsilon_\theta=30\%$)     |   87.4    |  7.0     | 9.0    |  87.5     |     |
>
>
> Table-reb 4: Ablation study on the safety threshold $\theta$ of barrier certificate. Larger $\theta$ indicates more critical safety needs.
>
> | Threshold $\theta$           | ACC@G | FPR@G | FNR@G | ARR@R |
> |------------------|:-----:|:-----:|:-----:|:-----:|
> | $0$     | 87.4      |  5.0     | 8.2  | 87.5      |
> | $-0.1$     | 84.0      |  4.2     | 12.5   |  87.5     |
> | $0.1$ |   90.4      |  5.6     | 6.5  | 87.5      |
> | $0.3$ | 86.9      | 7.4      | 4.2  |  87.5     |
>
>
>
> > evaluation in terms of the false negative rate
>
>
> We totally agree with the reviewer that false negatives are even worse than false positives, and we have already included the FNR metric in our above ablation results in Table-reb 2, 3, 4. We will also ensure to update all table results to include FNR in the revised version.
>
> > Appendix A.3. and B appear to be empty/missing.
>
> We enrich the supplementary material by presenting a list of more comprehensive examples in [this link](https://anonymous.4open.science/r/shieldagent-icml-rebuttal-30B0/rebuttal.pdf). Specifically, we provide (1) three examples of the extracted policy blocks and LTL rules in Figure-r. 4-6; (2) three examples of ASPM optimization in Figure-r. 7-9; (3) an example of the dataset sample in ShieldAgent-Bench in Figure-r. 10; (4) an end-to-end example of the shielding process in Figure-r. 11-16. We also fix the typos and provide a detailed algorithm for each optimization process in  Algorithm 1, 2, 3 of [this link](https://anonymous.4open.science/r/shieldagent-icml-rebuttal-30B0/figure_updates.pdf). We will ensure to add corresponding references and fix missing references in the updated version of our paper.
>
>
> > how often does your system fail because of a poorly tuned probabilistic circuit vs. a misformulated rule?
>
> We present detailed ablation results in Tables reb-2, 3, and 4 above. Regarding the PC weights, the results indicate that the guardrail’s performance remains comparably robust even under large perturbations (noise up to 30%), demonstrating the reliability of the probabilistic inference process within the robust ASPM policy model and the safety probability certification algorithm. Additionally, since the PC learns soft weights for each rule from real-world safety-related datasets paired with ground-truth labels, the impact of any misformulated rule is naturally mitigated, as such rules would receive very low weights during learning.
>
>
> > How do you set the hyperparameters for your method?
>
> Thank you for raising this insightful question! In our evaluation, we set the hyperparameters based on ablation experiments conducted on a held-out validation set of approximately 400 examples from ShieldAgent-Bench. For instance, we set the safety threshold to 0.1, as it achieved the best trade-off between guardrail accuracy and false positive rate compared to other candidate values.
>
>
> Once again, we sincerely thank the reviewer for their thoughtful feedback and for recognizing the contributions of our work for ensuring the safety deployment for LLM agents!

---

> > ### Comment · Reviewer_GLxp · 2025-04-03
> >
> > I have reviewed the authors' responses to the reviews. I am pleased to see the additional ablation studies, the inclusion of which in the paper will undoubtedly strengthen its conclusions.
> >
> > I will keep my current score. I do not feel confident raising it further as I do not believe myself sufficiently knowledgeable about other prior and concurrent work in this area, but I will reiterate that it seems like a very solid contribution to me and I would be happy to see the paper accepted.

---

> > > ### Author Response · Authors · 2025-04-06
> > >
> > > Dear Reviewer GLxp,
> > >
> > > We are sincerely grateful for your recognition of ShieldAgent’s contribution to ensuring the safety of LLM agents! Your valuable feedback has greatly helped us improve our work, and we deeply appreciate your support in recommending this paper for broader dissemination. Thank you very much!
> > >
> > > Submission 16287 Authors

---

### Official Review · Reviewer_sVmH · 2025-03-05

**Overall Recommendation:** 4

**Summary:**

Main findings:
- The paper introduces SHIELDAGENT, a novel guardrail agent designed for LLM agents to explicitly ensure compliance with safety policies during sequential decision-making through auto-mated probabilistic policy reasoning. It addresses significant vulnerabilities of LLMs to malicious instructions and attacks, which can lead to severe consequences such as privacy breaches and financial losses. Besides, this paper introduces a new benchmark, named SHIELDAGENT-WEB, which contains 2K safety-related instructions under two kinds of attacks.

Main results:
- This paper tests their algorithm on two benchmarks: SHIELDAGENT-WEB and ST-WEBAGENTBENCH. The SHIELDAGENT achieves SOTA results on both benchmarks under affordable inference costs.

Algorithmic ideas:
- The SHIELDAGENT consists of constructing a structured safety policy model and a shielding model.

-- Structured safety policy model: (1) uses GPT-4o to exhaust all the potential policies from the provided organization handbook which sets restrictions or guidelines based on four elements: definition, scope, policy description, and reference; (2) converts obtained policies to LTL structures with GPT-4o; (3) iteratively refines the verifiability of the policies and merges similar ones together; (4) trains the probabilistic circuit for policies and cluster similar circuits

-- Shield model: (1) selects the most related circuits and computes the safety probability; (2) shields the action with control barrier function (CBF) based on the safety probability; (3) provides the safety label, explanations, or violated rules based on CBF.

**Claims And Evidence:**

One claim assumes there exists policy document to do policy extraction. One problem may be how to deal with the case when there is not such document.

**Essential References Not Discussed:**

No, the key contribution is clearly itself with correctly cited related works.

**Experimental Designs Or Analyses:**

The “Direct” baseline seems to have smaller FPR with low accuracy in table 2. The main reason may be “Direct” tends to predict more “0” so that it may have high False Negative Rate (FNR). It is better to also show the FNR in the table.

**Methods And Evaluation Criteria:**

1.	Yes, this paper tests two datasets. One is created by themselves, and the other is an existing one.
2.	SHIELDAGENT outperforms existing baselines in both datasets.
3.	However, all benchmarks are related to web service. It would be better to add results in some other field.

**Other Comments Or Suggestions:**

1.	I don’t think GuardAgent, in section 2.2., focuses solely on textual space since GuardAgent also does experiments on a web service benchmark. However, I agree with the authors for the latter part, where GuardAgent just relies on the internal knowledge and reasoning ability to guard safety.
2.	In line 213-214, Page 4. Please correct the presentation “Prompts defined in Appendix C and Appendix C, respectively”.
3.	In table 2, the smallest overall FPR comes from “Direct” baseline, but the bold style is in the SHIELDAGENT.
4.	In Section 3.3 and 3.4, how to generate action shield plan and how to generate shielding code are not explained or with any examples.
5.	The Appendix part seems to be incomplete. Some titles do not have any content.

**Other Strengths And Weaknesses:**

1. Strengths
- This paper designs a novel pipeline to construct safety policy model, which can explicitly shield LLM agents.
- This paper provides a comprehensive benchmark, which contains 2k safety-related instructions across seven safety categories and six web environments under two kinds of attacks.

2. Weaknesses
- This paper only conducts experiments on Web services without other conditions.
- The appendix part seems to be incomplete.

**Questions For Authors:**

Q1: How to do pseudo training? Is the pseudo training high-time-cost since the agent trajectories need to be generated?

Q2: How to generate action shield plan and how to generate shielding code? Are there any explanations or any examples?

Q3: What is the action set in Figure 1? Is it the part of “Shielding Operations” in Section 3.3.

Q4: are the shielding operations and the toolbox general or task-specific? Are they the same for different kinds of tasks or needed to be fine-tuned or designed for other tasks?

**Relation To Broader Scientific Literature:**

1.	Safety: This paper introduces two general kinds of attacks, which are agent-based attacks and environment-based attacks. One contribution is that this paper considers both attacks in the design of SHIELDAGENT.
2.	Guardrails: Prior works mainly focus on guarding LLM models in natural languages or images rather than decision-making processes, like LlamaGuard, LlavaGuard, and SafeWatch. One method focuses on guarding LLM agents, named GuardAgent, mainly relies on model’s reasoning ability instead of explicitly shielding the policies of target LLM agent. Thus the main contribution of SHIELDAGENT is to achieve explicit safety for LLM agents in making sequential decisions.

**Theoretical Claims:**

N/A

---

> ### Author Rebuttal · Authors · 2025-04-01
>
> We sincerely appreciate the reviewer's recognition of our paper's novel contribution and thoughful suggestions!
>
> > how to deal with the case when there is not such document.
>
> Thank you for this insightful question! Instead of strictly requiring a document for policy extraction, we mainly aim to achieve explicit agent safety compliance against regulations and specifications defined in **natural language**, which reflects the vast majority of real-world cases. When no explicit documents are available, ShieldAgent can also leverage prompt engineering to extract implicit rules directly from data samples, as long as they can be described in natural language.
>
> > add results in some other field.
>
> To demonstrate the strong generalization capabilities of ShieldAgent in diverse agent tasks other than web services, we further evaluate it on two additional benchmarks, AgentHarm [1], which is a comprehensive agent risk benchmark beyond web agent tasks, and VWA-Adv [2], which involves diverse risk scenarios of GUI-based agents. We provide the detailed evaluation result of AgentHarm in Table-reb 5 below and defer the evaluation of VWA-Adv in Table-r. 2 of [this link](https://anonymous.4open.science/r/shieldagent-icml-rebuttal-30B0/rebuttal.pdf) due to space limit. The results demonstrate that ShieldAgent can effectively generalize and provide robust guardrails across different agent types, environments, and tasks.
>
> Table-reb 5: Guardrail performance comparison on **AgentHarm** across 11 harm categories.
>
> | Guardrail    | Fraud    | Cybercrime | Self-harm | Harassment | Sexual  | Copyright | Drugs   | Disinfo. | Hate    | Violence | Terrorism | Overall |
> |-------------|:-------:|:---------:|:--------:|:---------:|:------:|:--------:|:------:|:-------:|:------:|:-------:|:--------:|:-------:|
> | Direct      | 75.7 / 5.2 | 82.4 / **3.6** | 76.5 / **3.6** | 80.6 / 3.8  | 82.2 / **3.8** | 72.0 / 3.9  | **82.0** / 7.0 | 76.9 / 4.1 | 71.0 / **3.5** | 75.8 / 4.4  | 71.1 / 5.1  | 76.9 / 4.4  |
> | GuardAgent  | 82.6 / 4.7 | 66.1 / 4.0  | 75.1 / 4.5 | 75.9 / 3.4  | 82.1 / 6.3 | 69.6 / 4.3 | 76.6 / **3.8** | 80.1 / **3.2** | 77.7 / 3.7 | **92.4** / **3.3** | 83.9 / 4.2  | 78.4 / 4.1  |
> | **ShieldAgent** | **89.1** / **4.6** | **92.9** / 4.9 | **82.5** / 3.9 | **92.4** / **2.5** | **94.0** / 4.0 | **89.0** / **2.1** | 80.4 / 5.5 | **81.9** / 4.2 | **81.7** / 3.8 | 83.9 / 4.7 | **88.3** / **3.2** | **86.9** / **3.9** |
>
> > show the FNR in table 2
>
> We totally agree with the reviewer that the direct baseline achives lower FPR by predicting more "0". We will ensure to update all table results to include FNR in the revised version.
>
>
> > the supplementary material is too simple
>
> We enrich the supplementary material by presenting a list of more comprehensive examples in [this link](https://anonymous.4open.science/r/shieldagent-icml-rebuttal-30B0/rebuttal.pdf). Specifically, we provide (1) three examples of the extracted policy blocks and LTL rules in Figure-r. 4-6; (2) three examples of ASPM optimization in Figure-r. 7-9; (3) an example of the dataset sample in ShieldAgent-Bench in Figure-r. 10; (4) an end-to-end example of the shielding process in Figure-r. 11-16. We also fix the typos and provide a detailed algorithm for each optimization process in  Algorithm 1, 2, 3 of [this link](https://anonymous.4open.science/r/shieldagent-icml-rebuttal-30B0/figure_updates.pdf). We will ensure to add corresponding references and fix missing references in the updated version of our paper.
>
> > pipeline not explained with any examples.
>
> We provide an overview of the overall shielding procedure in Figure-r. 11, and provide an end-to-end example of the shielding plan generation process in Figure-r. 12-14 in [this link](https://anonymous.4open.science/r/shieldagent-icml-rebuttal-30B0/rebuttal.pdf). Additionally, we provide a detailed example illustrating the shielding code generation and verification process in Figure.-r 15, and also illustrate the safety probability inference process in Figure.-r 16.
>
> > How to do pseudo training?
>
> We follow [3] to conduct pseudo training, which actually lowers the time and data costs by leveraging some heuristics to simulate the labels of the training data. However, it slightly compromises performance as indicated by the results in Table-reb 3 in our response to reviewer oQJ8.
>
> > What is the action set in Figure 1?
>
> Yes, the action set contains all the shielding operations which can be further extended to meet diverse guardrail requirements.
>
> > are the shielding operations and the toolbox general or task-specific?
>
> While in our experiments, we leverage the same built-in tools to ensure a fair and controllable evaluation, the tool library could be easily extended to include diverse new tools and specialized operations since our agent is built following the MCP protocol (as shown in Figure-r. 11-16).
>
> [3] Kang, M., & Li, B. R2-Guard: Robust Reasoning Enabled LLM Guardrail via Knowledge-Enhanced Logical Reasoning. ICLR 2025

---

> > ### Comment · Reviewer_sVmH · 2025-04-02
> >
> > The authors addressed my concerns and also provided additional experiments. I am glad to raise my recommendation to "accept".

---

> > > ### Author Response · Authors · 2025-04-07
> > >
> > > Dear Reviewer sVmH,
> > >
> > > We are very glad that our response has addressed your concerns! We sincerely appreciate your recommendation to accept our work, and we are excited to contribute ShieldAgent to the community’s efforts toward building safer and more reliable LLM agents. Thank you!
> > >
> > > Sincerely,
> > >
> > > Authors of Submission 16287

---

### Official Review · Reviewer_oQJ8 · 2025-03-10

**Overall Recommendation:** 3

**Summary:**

This paper presents ShieldAgent, a new technique for determining whether LLM
outputs conform to a given policy. ShieldAgent starts by using an LLM to
formalize a policy document and produce a set of rules expressed in LTL. These
LTL formulae are embedded into probabilistic circuits which can be used to
efficiently estimate the safety of potential actions. These probabilistic
circuits can also be used to construct a barrier function which ensures the
safety of an LLM agent over a long sequence of decisions. The paper proposes a
new dataset designed to test the efficacy of guardrails for LLM agents. On both
this new dataset and a preexisting dataset, ShieldAgent outperforms prior work.

**Claims And Evidence:**

The main claim of the paper, that ShieldAgent works better as a guardrail for
LLM agents, is well supported with experiments. Across several different risk
categories, ShieldAgent is almost universally more accurate than prior methods
at identifying potentially unsafe behaviors. The practical impact of this
accuracy is shown with additional experiments that use ShieldAgent on a set of
internet interaction tasks. These experiments show that ShieldAgent results in
safer behavior compared to prior guardrails.

**Essential References Not Discussed:**

To the best of my knowledge, the authors discuss all essential relevant
literature.

**Experimental Designs Or Analyses:**

The experiments are generally well-designed. They could be improved with the
inclusion of some ablation studies. ShieldAgent includes so many interacting
components that it is difficult for me to tell what the impacts of different
pieces of the system are. For example, it would be helpful to understand the
costs of omitting the model checker or the KV-cache.

**Methods And Evaluation Criteria:**

The datasets used for evaluation are appropriate and include both an established
dataset and a novel dataset designed for the use case targeted by the proposed
technique. The experiments include both prior work in this setting as well as a
few plausible baseline techniques to provide evidence that ShieldAgent is a
useful approach.

**Other Comments Or Suggestions:**

The paper mentions that many external tools are used by ShieldAgent but gives
little detail about what those tools are or how they are used. For example,
Section 3.3 says that ShieldAgent uses a formal verification component that
"conducts model checking to rigorously validate rule compliance". But model
checking can refer to quite a few different techniques, so it would be useful to
include some details on how that model checking is conducted. As another
example, 3.2.3 describes "probabilistic safety inference" at a very high level,
but it's not clear to me how such a computation would actually be carried out.

At the same time, there are some places in the paper where space is used to
explain fairly standard concepts. For example, in 3.2.3, there is a definition
of the binary cross-entropy loss. In my opinion, this definition is so standard
that it could be removed or moved to the appendix in order to save some space
for the more novel details of this paper.

**Other Strengths And Weaknesses:**

The main strength of the paper in my opinion is the experimental results. They
are extensive and show an impressive improvement over prior work in terms of
accurately classifying potentially unsafe behavior.

The main weakness of the paper which is not captured in the other fields of this review is the
number of interacting components. This is not inherently a problem, but it makes
it difficult to include enough detail to really understand the proposed
technique in the paper. See the "other comments or suggestions" section below
for a few specific places where I felt this detracted from the paper.

**Questions For Authors:**

I'm not quite sure I understand the definition of $P_S(a \mid o)$. As I
understand it, it is related to the probability of picking action $a$
(denoted $P(a \mid o)$) as well as "the safety probability when all rules are
satisfied", $P_{ub}(o)$. First, why is it the case that when all rules are
satisfied, the probability of safety is not one? Second, I'm not sure why the
probability of safety is defined with respect to the probability of picking
action $a$ rather than the probability that the system remains safe after taking
action $a$.

Second, could the authors elaborate on the soundness of the barrier certificate?
It seems to me that the second requirement,
$P_S(a \mid o) - P_s(o) \le \epsilon_{tol}$, allows the probability of unsafe
behavior to increase by $\epsilon_{tol}$ with each time step. Doesn't this allow
the system to behave unsafely over long time horizons?

**Relation To Broader Scientific Literature:**

The related work section situates this work well within related literature.
Because LLM's are often used sequentially, it is important to ensure they are
safe even when used repeatedly to generate a sequence of decisions.

**Theoretical Claims:**

This paper does not make any novel theoretical claims.

---

> ### Author Rebuttal · Authors · 2025-04-01
>
> We really appreciate the reviewer's valuable suggestions and we have accordingly updated the paper to include additional results and examples in [this link](https://anonymous.4open.science/r/shieldagent-icml-rebuttal-30B0/rebuttal.pdf).
>
> > inclusion of some ablation studies
>
> We provide a detailed ablation study regarding the different components, different PC weights, and different safety thresholds in Table-reb 2, 3, 4 below. The results show that guardrail performance and efficiency are primarily influenced by tool calling and model-checking components, which are critical for detecting rule violations. Additionally, we observe that PC significantly reduces the FPR and remains stable even under large perturbations, demonstrating the robustness of our safety certification algorithm.
>
> Table-reb 2: Ablation study on different components
>
> | Component           | ACC@G | FPR@G | FNR@G | ARR@R | Time |
> |------------------|:-----:|:-----:|:-----:|:-----:|:-----:|
> | w/o Model checking     |85.4      |  7.2     | 9.0    | 80.9      | 24.0 |
> | w/o PC     | 82.5      |  13.0     | 2.3    | 87.5      | 31.1      |
> | w/o History cache | 90.4      |  5.6      | 6.5   | 87.5      | 42.0     |
> | w/o Workflow memory  | 86.0      |  7.0      | 8.5   | 82.0      | 38.2     |
> | w/o Tool calling          | 81.4      | 6.2      | 11.3      |  71.3     | 48.9     |
>
> Table-reb 3: Ablation study on PC weights.
>
> | Method           | ACC@G | FPR@G | FNR@G | ARR@R |
> |------------------|:-----:|:-----:|:-----:|:-----:|
> | Learn from real data      | 90.4      |  5.6      | 6.5   | 87.5      |
> | Pseudo-learning |  88.0     |  6.7     | 8.5    | 87.5      |
> | FOL ($\theta \to \infty$)     | 82.5      |  13.0     | 2.3    | 87.5      |
> | Perturbed ($\epsilon_\theta=10\%$)     |  89.7     |  5.9     |  7.7   |   87.5    |
> | Perturbed ($\epsilon_\theta=30\%$)     |   87.4    |  7.0     | 9.0    |  87.5     |     |
>
>
> Table-reb 4: Ablation study on the safety threshold $\theta$ of barrier certificate. Larger $\theta$ indicates more critical safety needs.
>
> | Threshold $\theta$           | ACC@G | FPR@G | FNR@G | ARR@R |
> |------------------|:-----:|:-----:|:-----:|:-----:|
> | $0$     | 87.4      |  5.0     | 8.2  | 87.5      |
> | $-0.1$     | 84.0      |  4.2     | 12.5   |  87.5     |
> | $0.1$ |   90.4      |  5.6     | 6.5  | 87.5      |
> | $0.3$ | 86.9      | 7.4      | 4.2  |  87.5     |
>
>
> > the supplementary material is unfinished
>
> We enrich the supplementary material by presenting a list of more comprehensive examples in [this link](https://anonymous.4open.science/r/shieldagent-icml-rebuttal-30B0/rebuttal.pdf). Specifically, we provide (1) three examples of the extracted policy blocks and LTL rules in Figure-r. 4-6; (2) three examples of ASPM optimization in Figure-r. 7-9; (3) an example of the dataset sample in ShieldAgent-Bench in Figure-r. 10; (4) an end-to-end example of the shielding process in Figure-r. 11-16. We also fix the typos and provide a detailed algorithm for each optimization process in  Algorithm 1, 2, 3 of [this link](https://anonymous.4open.science/r/shieldagent-icml-rebuttal-30B0/figure_updates.pdf).
>
>
> > tools and examples used
>
> We provide a comprehensive example of the tools used by ShieldAgent based on MCP protocol in Figure-r. 11 and explain step-by-step the different tool calling purposes within the shielding plan in Figure-r. 11-16. Specifically, we leverage Prover9/Mace4 as model checking tool and provide a detailed example of it in Figure.-r 15. We also illustrate the safety inference process in Figure.-r 16.
>
>
> > paper writing arrangement
>
> We will ensure to follow the reviewer's suggestion and update the paper presentation to focus more on delivering the key contributions of our method.
>
>
>
> > clarification on $P_S(a\mid o)$
>
> We adopt MLN to model safety probability w.r.t. explicit rule compliance, i.e., $P(\mu)=\frac{1}{Z}\exp(\sum_{r \in R}\theta [\mu \sim r])$, where $z$ is a possible assignment of predicates, and $Z$ is the partition function which sums up all possible assignments. Thus even if all rules are satisfied, the probability of safety only reaches an upper bound instead of approaching one. And since the action predicate $a$ is also part of the MLN state space, $P(a \mid o)$ essentially denotes the probability that the system remains safe after taking action $a$.
>
> > clarification on barrier certificate
>
> We thank the reviewer for noting this typo and we apologize for the confusion. The condition $P_s(a\mid o)-P_s(o)\geq \epsilon$ only applies when the safety probability $P_s(o)$ is smaller than a prescribed threshold, i.e., outside the tolerable safety region. In such cases, the condition enforces that the safety probability must increase by at least \epsilon at each step. Therefore, rather than allowing the probability of unsafe behavior to accumulate, this condition ensures that the system progressively becomes safer when it is in an unsafe state.

---

### Official Review · Reviewer_tHcj · 2025-03-13

**Overall Recommendation:** 2

**Summary:**

This paper proposes SHIELDAGENT, an LLM-based guardrail agent, to enforce explicit safety policy compliance of the action sequences of other LLM agents via automated probabilistic reasoning. SHIELDAGENT constructs an action-based probabilistic safety model (APSM) by extracting verifiable rules from policy documents, refining and clustering them into action-conditioned probabilistic circuits, and learning rule weights. During inference, it localizes relevant rule circuits, generates shielding plans, and performs probabilistic inference to assign safety labels and report violations. To evaluate it, the SHIELDAGENT-WEB dataset is introduced, which contains 2K safety-related instructions across various risk categories and web environments with paired risky trajectories. Experiments show that SHIELDAGENT achieves state-of-the-art performance on SHIELDAGENT-WEB and existing benchmarks.

**Claims And Evidence:**

The methodology and evaluations presented in this paper lack important running examples for readers to understand.

Please refer to Weaknesses for more details.

**Essential References Not Discussed:**

Please see relation to broader literature above.

**Experimental Designs Or Analyses:**

In this paper, the authors constructed a dataset, named ShieldAgent-Web, consisting of AgentPoison and AdvWeb attacks to evaluate the robustness of the proposed ShieldAgent framework.

However, no sample in the dataset has been given in the paper or the supplementary material, making it hard to understand to what extent the proposed dataset can represent threats against agent systems in real-world applications.

Please refer to Weaknesses for more details.

**Methods And Evaluation Criteria:**

The methods proposed in the paper is relevant to safeguarding LLM agents, which face the challenge of security threats in simulated environments. SHIELDAGENT's approach of constructing an action-based probabilistic safety model (APSM) seems to be a working solution compared with traditional static approaches, and the four key shielding operations of SHIELDAGENT work in concert with the APSM.

Please refer to Weaknesses for more details.

**Other Comments Or Suggestions:**

Please refer to weaknesses and questions.

**Other Strengths And Weaknesses:**

### Strengths

- SHIELDAGENT offers a comprehensive approach to safeguarding LLM agents.
- The SHIELDAGENT-WEB dataset is proposed for evaluation.

### Weaknesses

- The presentation of this paper can be improved.
  - In terms of the methodology, the overall framework is an RAG-based linear classifier which uses safety rules extracted by LLMs to ensure safe operations of LLM-based agents. The authors put much effort in modeling the linear classifier with the term "Action-based Probabilistic Safety Model", but did not provide essential examples of the rules and the verification and pruning process. It is also not clear how the policy optimization contribute to the safety performance.
  - In terms of the figures, for instance, there are too many elements in Figure 1 that the fonts of each text is small to recognize. The caption of the figure also does not provide enough substantial information to help me understand what operations are to be performed.
- The evaluation of the paper is confusing, maybe questionable.
  - No sample has been provided in the ShieldAgent-Web dataset. It is hard to imagine how realistic are the attacks considered in the paper. It is suggested that the dataset should be open-sourced, or at least typical samples should be provided for reviewing.
  - Although SHIELDAGENT performs well on the datasets used in the experiments, its generalization to new and unseen scenarios may be limited. The evaluation on diverse scenarios is expected.

**Questions For Authors:**

- Could you provide a more detailed explanation of Figure 1(Top)? Now that rules are clustered in the Redundancy Pruning process, but the graph shows that it is clustering the conditional predicates? Moreover, what does the arrow from one grey circle to another represent? Does it mean the split of a rule into two rules, or a logical context between the two rules? It seems that its meaning in Redundancy Pruning and Action-based Probabilistic Circuit is different. A clearer explanation to the figure and these questions would be very helpful.
- In the Probabilistic-Constrained Safety Certification part, is the condition $P_s(a∣o)-P_s(o)<\epsilon_{tol}$ a necessary and sufficient condition for action safety? Could you provide more theoretical basis or experimental evidence?
- In the baselines, why Rule Traverse will exhibit high FPR? Sequentially verifying rules one by one seems to have a low FPR result. Could you explain this baseline in more detail?
- As stated in Sec. 3.2.1, the policy blocks are extracted by GPT-4o. How do the authors ensure that rules extracted by LLMs align with human values?

**Relation To Broader Scientific Literature:**

- The composition of the probabilistic safety certification module highly resembles [1], [2]. It is suggested that the authors can discuss the relationship to these works and the key contribution of this work.

[1] Knowledge Enhanced Machine Learning Pipeline Against Diverse Adversarial Attacks. ICML 2021.

[2] Improving Certified Robustness Via Statistical Learning with Logical Reasoning. NeurIPS 2022.

**Theoretical Claims:**

In terms of the safety certification, the use of control barrier function (CBF)-inspired conditions $|P_s(a∣o)|<\epsilon_{tol}$ and $P_s(a∣o)-P_{s}(o)<\epsilon_{tol}$ to determine action safety seems to be a valid approach.

Despite the multiple equations used to model the policy model decision process, it seems to be a linear classifier based on the rules extracted. Since no rigorous proof or theory has been provided in the paper, I take this certification as empirical.

---

> ### Author Rebuttal · Authors · 2025-04-01
>
> Thank you for your valuable feedback! We have followed your suggestions and improved our paper to incorporate more examples and additional experiment results.
>
>
> > lack examples
>
> We provide a list of more comprehensive examples in [this link](https://anonymous.4open.science/r/shieldagent-icml-rebuttal-30B0/rebuttal.pdf). Specifically, we provide (1) three examples of the extracted policy blocks and LTL rules in Figure-r. 4-6; (2) three examples of ASPM optimization in Figure-r. 7-9; (3) an example of the dataset sample in ShieldAgent-Bench in Figure-r. 10; (4) an overview of the overall shielding procedure in Figure-r. 11; (5) an end-to-end example of the shielding plan generation process in Figure-r. 12-16.
>
>
> > no sample of the dataset
>
> We include an example from ShieldAgent-Bench in Figure-r.10 and a comprehensive comparison with existing datasets in Table-r. 4. To ensure that our dataset represents agent threats in real-world applications, we construct it by attacking real-world SOTA agents with practical security-related targets to elicit unsafe trajectories and conduct thorough human review to ensure quality. To ensure transparency and ease of verification, we manually annotate the potential safety violations of each agent action step.
>
> >  missing references in the algorithm and typos.
>
> We have updated the paper to fix all missing references and typos, and also provided the updated versions in Algorithm 1, 2, 3 of [this link](https://anonymous.4open.science/r/shieldagent-icml-rebuttal-30B0/figure_updates.pdf).
>
> > key contribution of the probabilistic safety certification module
>
> Our key contribution on top of [1,2] is that we are the first to successfully (1) scale up the probabilistic verification process for a very large predicate space by clustering it by different actions and constructing local action-based circuits; (2) extend the static knowledge rules in previous works to temporal logics and integrate it with barrier certificates to facilitate verifying agent action sequence.
>
> > examples of the rules and the verification and pruning process.
>
> We present detailed statistics of the number of rules, predicates, and their vagueness score (evaluated by GPT-4o-as-judge) in Table-r. 1 and Figure-r. 1-3; examples of rules during the optimization process are presented in Figure-r. 7-9.
>
> > Update Figure 1 and caption
>
> We have updated Figure 1 and its captions in Figure-r. 1 of [this link](https://anonymous.4open.science/r/shieldagent-icml-rebuttal-30B0/figure_updates.pdf). Due to limited space, we defer a more detailed illustration of the shielding process in Figure-r. 11 and explain step-by-step the shielding plan generation process in Figure-r. 11-16.
>
>
> > evaluation on diverse scenarios is expected.
>
> To demonstrate the strong generalization capabilities of ShieldAgent to diverse agent tasks and unseen scenarios, we further evaluate it on two additional benchmarks, i.e. VWA-Adv [1] (diverse risks of vision-based web agents), and AgentHarm [2], which is a comprehensive agent risk benchmark (not limited to web agent) that involve 11 unseen scenarios not present in our dataset. We provide a summary of the result in Table-reb 1 below and defer the detailed results in Table-r. 2 and Table-r, 3 of [this link](https://anonymous.4open.science/r/shieldagent-icml-rebuttal-30B0/figure_updates.pdf). The results demonstrate that ShieldAgent can generalize well to unseen domains and provide robust guardrails across different agent types, environments, and tasks.
>
> Table-reb 1: Comparison of guardrails on VWA and AgentHarm.
>
> | Guardrail     | VWA-Adv (ACC ↑ / FPR ↓) | AgentHarm (ACC ↑ / FPR ↓) |
> |--------------|:----------------------:|:------------------------:|
> | Direct      | 90.3 / 4.2            | 76.9 / 4.4              |
> | GuardAgent  | 89.9 / 4.4            | 78.4 / 4.1              |
> | **ShieldAgent** | **94.1 / 3.4**        | **86.9 / 3.9**          |
>
>
> > Explain Figure 1 top
>
> The grey arrow in Figure 1 denotes logic connections. VR splits a compound rule into two atomic sub-rules, and RP clusters the rule space by semantic similarity to reduce redundancy. After optimization, ASPM represents them as Laplace matrix and applies spectral clustering to obtain rule circuits.
>
> > is the condition necessary and sufficient
>
> The condition is a sufficient condition to asymptotically guarantee safety based on MDP assumption. We will include a detailed proof in the updated version of the paper.
>
> > high FPR of Rule Traverse?
>
> We found verifying rules one by one usually results in the guardrail being over-cautious and thus induces a higher FPR.
>
>
> > rules align with human values?
>
> We guarantee the accuracy and quality of the rules by manually validating them w.r.t. the document source which is preserved during optimization.
>
>
> [1] Wu, C. et al. Dissecting Adversarial Robustness of Multimodal LM Agents. ICLR 2025
>
> [2] Andriushchenko, M., et al. Agentharm: A benchmark for measuring harmfulness of llm agents. arXiv preprint

---

> > ### Comment · Reviewer_tHcj · 2025-04-04
> >
> > Thanks for the authors' detailed response. After reading the rebuttal, some of my concerns have been addressed. However, I still have the following concerns regarding the paper.
> >
> > Firstly, the presentation of the original paper can be largely improved. The overall working mechanism of the proposed ShieldAgent framework remain unclear until I review the samples provided in the rebuttal materials. It is suggested that the theories proposed in the paper should guarantee the safety of the procedure instead of causing difficulties to the understanding of readers.
> >
> > Secondly, the quality of the extracted rules is doubtful. No public access to the generated rules database is provided in the paper and the rebuttal except several examples. As stated in the rebuttal, no quantitative evaluation has been conducted regarding the quality and alignment of the rules to human values as well. The authors claim that one of the main contribution of the paper is scaling up the verification process, but they manually validate the rules.
> >
> > For the reasons above, I will keep my score. I hope the ACs take all the reviewers' perspectives into accout and make the final recommendation.

---

> > > ### Author Response · Authors · 2025-04-06
> > >
> > > Dear Reviewer tHcj,
> > >
> > > We are glad that our response has addressed your previous concerns, and we would like to provide additional details in light of your follow-up questions!
> > >
> > > > presentation of the original paper
> > >
> > > Thank you for your suggestions, following which we have largely improved the paper's presentation and summarized our updates as follows (detailed in [[this link](https://anonymous.4open.science/r/shieldagent-icml-rebuttal-30B0/rebuttal.pdf)]). We will integrate all discussions in the rebuttal into the final version.
> > >
> > > + **Additional Case Studies**: We added various running examples in the appendix to explain each component of ShieldAgent: a) three examples of the extracted policy blocks and LTL rules in Figure-r. 4-6; b) three examples of ASPM optimization in Figure-r. 7-9; c) an overview of the overall shielding procedure in Figure-r. 11; d) an end-to-end example of the shielding plan in Figure-r. 12-16.
> > >
> > > + **Quantitative Analysis of Policy Optimization**: We provided additional statistics of the rule extraction process in Table-r. 1 and Figure-r. 1-3, as well as a detailed human alignment analysis in Table-reb 2-3 below.
> > >
> > > + **Dataset Quality Demonstration**: We compared ShieldAgent-Bench with previous works in Table-r. 4 and provided an example in Figure-r. 10, showing that it effectively captures the security threats of real-world agent applications.
> > >
> > > + **Clarified Methodology**: We further clarified our safety certification methodology and provided rigorous theoretical proof to demonstrate that it effectively guarantees agent safety via barrier certificates. Due to rebuttal constraints, we will ensure to disclose this proof in the camera-ready version.
> > >
> > > + **Revised Figures and Corrections** We updated Figure 1 and its captions, fixed all typos and references, and provided detailed pseudocode for each optimization process in Algorithm 1, 2, 3 in [this link](https://anonymous.4open.science/r/shieldagent-icml-rebuttal-30B0/figure_updates.pdf).
> > >
> > > + **Additional Evaluation**: We evaluated ShieldAgent on two diverse agent safety benchmarks VWA-Adv and AgentHarm and provide the results in Table-r. 2-3.
> > >
> > > We sincerely hope these updates have enhanced the clarity of our paper and can help readers better understand our work.
> > >
> > > > scalability of rule extraction
> > >
> > > We really appreciate the reviewer's thoughtful question!
> > >
> > > We would like to first clarify that since the number of policies is finite (e.g. GitLab Policies, OpenAI Use Policies, EU AI Act), our automatic rule extraction pipeline requires only a **one-time, cost-efficient human-in-the-loop verification process which can be performed completely offline, once and for all for each policy**. And the resulting rule database can then be applied generically across various agents, enabling us to further scale up the verification process through the proposed automatic action tree-based probabilistic circuit and retrieval-based verification method.
> > >
> > > Following the reviewer's suggestions, we would like to also provide additional evidence that our automatic rule extraction pipeline is **scalable in terms of human efforts** and capable of producing rules that **strongly align with human intent**. As shown in the verification statistics in Table-reb 2, **fewer than 10% of the rules extracted by our pipeline require manual correction**, while the remaining 90% already align well with human values and require no modification, keeping total human verification time under 1 hour across different environments. Compared to traditional policy-based guardrail that requires lengthy human curation, our automatic rule extraction pipeline can efficiently produce accurate, large-scale, and human-aligned rules, substantially reducing human efforts and achieve scalable rule extraction.
> > >
> > > Table-reb 2: Human verification statistics for extracted rules. We report the total number of extracted rules, manually corrected rules, the update ratio, and the total human verification time.
> > >
> > > | Environment     | #Total Rules | #Updated Rules    |  #Update Ratio (\%) | #Human Working Hours |
> > > |--------------|:---------------:|:-------------------:|:-------------------:|:-------------------:|
> > > | Shopping      |   240        |    27          |   11.3    | $\leq 1h$            |
> > > | CMS  |    120              | 10      | 8.3            |   $\leq 0.5h$          |
> > > | Reddit|     178          | 13       |     7.3       |   $\leq 1h$          |
> > > | GitLab|      198              | 23    |    11.6         |   $\leq 1h$          |
> > > | Maps|        104        | 5        |  4.8           |  $\leq 0.5h$           |
> > > | SuiteCRM |   240        |    12          | 5.0            |  $\leq 1h$           |
> > >
> > > Once again, we sincerely thank the reviewer for all the insightful feedback, and we would be truly grateful if you would kindly consider revisiting the score. As also recommended by other reviewers, we believe ShieldAgent could contribute meaningfully to the community in building safer and more capable agents. Thank you!

---

### Decision · Program_Chairs · 2025-05-01

**Decision:**

Accept (poster)

**Comment:**

This paper proposes ShieldAgent, a framework for providing safety to LLM agents. The frameworks includes steps such as refining the policy documents as verifiable rules, policy model structure optimization, and probabilistic safety certification.

The authors provided extra results during the rebuttal, which addressed some of the reviewers' concerns, and the paper meets the ICML standards. They should address the rest of the concerns in the final version.